# The Role of Adenosine in γδ T-Cell Regulation of Th17 Responses in Experimental Autoimmune Uveitis

**DOI:** 10.3390/biom13101432

**Published:** 2023-09-22

**Authors:** Hui Shao, Henry J. Kaplan, Deming Sun

**Affiliations:** 1Department of Ophthalmology and Visual Sciences, Kentucky Lions Eye Center, University of Louisville, Louisville, KY 40202, USA; 2Department of Ophthalmology and Biochemistry & Molecular Biology, St. Louis University School of Medicine, Saint Louis, MO 63104, USA; 3Doheny Eye Institute and Department of Ophthalmology, David Geffen School of Medicine at UCLA, Los Angeles, CA 90033, USA

**Keywords:** adenosine deaminase (ADA), autoimmunity, adenosine receptor (AR), adenosine triphosphate (ATP), adenosine 2A receptor (A2AR), experimental autoimmune uveitis (EAU), gamma delta (γδ) T cells, regulatory T cells, Th17 cells, uveitis

## Abstract

Autoimmune diseases caused by T cells can arise from either T-helper 1 (Th1) or T-helper 17 (Th17)-type pathogenic T cells. However, it is unclear whether these two T-cell subsets are influenced by distinct pathogenic factors and whether treatments that are effective for Th1 responses also work for Th17 responses. To compare these two pathogenic responses, we conducted a systematic analysis in a mouse model of experimental autoimmune uveitis (EAU) to identify the factors that promote or inhibit each response and to determine their responses to various treatments. Our study found that the two types of pathogenic responses differ significantly in their pathological progressions and susceptibility to treatments. Specifically, we observed that extracellular adenosine is a crucial pathogenic molecule involved in the pathogenicity of inflammation and T-cell reactivity and that reciprocal interaction between adenosine and gamma delta (γδ) T cells plays a significant role in amplifying Th17 responses in the development of autoimmune diseases. The potential effect of targeting adenosine or adenosine receptors is analyzed regarding whether such targeting constitutes an effective approach to modulating both γδ T-cell responses and the pathogenic Th17 responses in autoimmune diseases.

## 1. Introduction

Over the past three decades, substantial evidence has emerged to support the notion that the major pathogenic T-cell subset in autoimmune diseases produces interferon gamma (IFN-γ). This cell subset was identified as containing Th1 pathogenic T cells [1,2,3,4]. Recently, various studies have demonstrated that T-cell subsets producing IL-17 are critically involved in disease pathogenesis as well [5,6,7]. However, two issues remain largely unexplored: whether Th1 and Th17 pathogenic T cells are induced by different pathogenic or environmental factors and whether they respond differently to the same treatment. Clarifying these lingering questions has significant potential to improve the treatment of diseases caused by Th1 and/or Th17 pathogenic responses.

## 2. Adenosine and Inflammation

Adenosine plays a critical role in the pathophysiological changes of diseases, particularly inflammatory diseases [8,9,10,11]. Studies have demonstrated that adenosine can have direct effects on T cells, inhibiting effector T-cell responses [12,13] but promoting the differentiation and expansion of regulatory T cells (Tregs) [14,15]. Adenosine also regulates dendritic cell (DC) function, promoting tolerogenic DCs that induce T-cell tolerance and inhibit autoimmune responses [12,16]. Various immune cells serve as rich sources of adenosine, including B cells [17], neutrophils [18], mast cells [19], endothelial cells [18,20], and T cells [10]. Adenosine generated in injured tissue is destroyed by enzymes, mainly adenosine deaminase (ADA).

Extracellular adenosine levels are low in healthy individuals [21] but increase 100- to 1000-fold during tissue injury and inflammation [10,22] due to the amplified release of adenosine triphosphate (ATP) from activated and injured cells into extracellular space, followed by its dephosphorylation to ADP and AMP and, finally, its conversion of AMP to adenosine [23]. This final step is catalyzed by the extracellular enzyme CD73. ATP augments immune responses when it leaks into the extracellular compartment during inflammation [24,25], and the degraded ATP metabolite adenosine is profoundly anti-inflammatory [26,27]. Overall, adenosine is a critical modulator of immune responses and a promising target for therapeutic interventions in a range of inflammatory and autoimmune diseases. The discovery of adenosine’s effect on inflammation and immune responses has inspired attempts to treat immune dysfunctions by targeting the adenosine receptor (AR) [9,28,29].

While the influence of adenosine on numerous immune responses has been documented, previous studies examining the effect of adenosine on T-cell responses have focused mainly on the Th1 cells. These studies have demonstrated that adenosine is inhibitory [30,31,32], though its effects on the recently discovered Th17 responses remain largely unexplored. This also applies to the role of adenosine in γδ T cells, which are important regulators of Th17 responses [30,33]. To determine whether adenosine has a similar effect on Th17 T cells, we employed a well-established model of experimental autoimmune uveitis (EAU), which resembles human autoimmune posterior- or pan-uveitis, and studied this model in both in vitro and in vivo parallel Th1 and Th17 responses. Our experiments showed that the net effect of adenosine on Th17 responses is augmenting, in marked contrast to its inhibitory effect on Th1 responses [32,34]. We were surprised to observe that alpha beta (αβ) and γδ T cells differ significantly in their responses to adenosine and that adenosine enhances γδ T-cell activation but inhibits αβ T-cell activation [32,34].

Our recent studies have found that adenosine plays a crucial role in the function of γδ T cells that regulate Th17 responses. It is widely accepted that Treg cells promote self-tolerance by suppressing undesirable immune responses; tightly controlled Treg cells are essential for shaping the desired immune responses, and control of these cells can effectively manipulate autoimmunity [35,36,37]. The stability and function of Treg cells have been shown to be critical in the maintenance of peripheral immune tolerance. While previous studies have shown that Foxp3^+^ Treg cells are effective in regulating Th1 responses and maintaining peripheral tolerance, their effectiveness in regulating Th17 responses has been a controversial topic.

Previous studies have also shown that γδ T cells play an important role in regulating immune responses [38,39]. Our studies have demonstrated that γδ T cells are essential for the adenosine-mediated suppression of Th17 responses and that adenosine signaling in γδ T cells plays a key role in the immunoregulatory process [40,41,42]. Our studies have found that γδ T cells have a substantial regulatory effect on Th17 response [33,43], while the effect of Foxp3^+^ Treg cells on Th17 responses was modest, suggesting that γδ T cells may have an important function in maintaining immune balance and preventing autoimmune diseases. Our studies indicate that adenosine signaling in γδ T cells plays a critical role in regulating immune responses and maintaining peripheral immune tolerance and that targeting adenosine signaling in γδ T cells may represent a promising therapeutic strategy for autoimmune and inflammatory diseases.

## 3. Adenosine Effect in γδ T Cell Activation

ATP is well known for its function as a universal energy currency. Interestingly, ATP plays a completely different role in the extracellular compartment, where it functions as a signaling molecule through the activation of nucleotide receptors [44]. Inflammatory conditions are associated with the extracellular release of nucleotides, particularly ATP. In the extracellular compartment, ATP predominantly functions as a signaling molecule through the activation of nucleotide receptors referred to as purinergic P2 receptors [22,44], which are expressed through a wide range of different tissues, implicating their role in innate or adaptive immune responses [22,45,46]. The significance of extracellular adenosine 5′-triphosphate (eATP) in cell-to-cell communication in the nervous and vascular systems has been thoroughly studied for years, but its role in the immune system is less well known.

γδ T cell is mainly activated in the periphery. Vγ1+ and Vγ4+ are the major subsets of the peripheral γδ T cells. Both subsets respond to adenosine. Since in immunized mice, Vγ4+ γδ T cells are the dominant subset, and as we previously reported that Vγ1+ and Vγ4+ T cells have distinct activation requirements, a greater effect of adenosine on γδ T cells in immunized mice should be contributed by the Vγ4+ cells. Our studies showed that Th17 responses are compromised under conditions in which γδ T cells are functionally defective (TCR-δ^−/−^ mouse); however, restoration of the Th17 responses was only observed when TCR-δ^−/−^ mice were administered with activated, but not resting, γδ T cells [31]. Therefore, investigations regarding how γδ T cells are activated in pathogenic processes and how this activation affects their pro- and anti-inflammatory activity should help unravel the mechanism of Th17-dependent autoimmunity. Many factors, including cytokines and Toll-like receptor (TLR) ligands, can activate γδ T cells in the absence of TCR ligation and increase their pro-inflammatory effect [31,47,48,49]. In our study on the role of adenosine in γδ T cell-mediated immunoregulation, we found that adenosine is critical for γδ T-cell activation and regulation and that the blockade of adenosine receptors (ARs) on γδ T cells greatly reduced γδ T-cell activation. Furthermore, γδ T cells obtained from A2AR-deficient mice could not be fully activated; therefore, their enhancing effect on Th17 responses was greatly diminished. Additionally, adenosine acts on γδ T cells via the A2AR, which is highly expressed in these cells [32,50]. In these studies, we found that A2AR signaling in γδ T cells enhances their pro-inflammatory effect by promoting their proliferation, cytokine production, and the activation of antigen-presenting cells (APCs). Moreover, we demonstrated that A2AR activation on γδ T cells increases their migration to inflammatory sites, further promoting their regulatory effect. These findings suggest that adenosine/A2AR signaling is critical for γδ T-cell-mediated immunoregulation and that targeting this pathway may provide new therapeutic opportunities for autoimmune diseases. This insight highlights the importance of maintaining a balance between adenosine levels and γδ T-cell activation in order to avoid excessive Th17 responses and the development of autoimmune diseases. Further research is needed to fully understand the complex interplay between adenosine, γδ T cells, and Th17 responses and to identify potential therapeutic targets for regulating this pathway in autoimmune diseases.

## 4. γδ T Cell Function Correlates to Their Expression of CD73

CD73 is a glycosyl phosphatidylinositol-linked membrane protein that is pivotal in the conversion of immunostimulatory ATP into immunosuppressive adenosine [51,52]. It has been viewed as an immunological switch that shifts ATP-driven pro-inflammatory activity toward an anti-inflammatory state mediated by adenosine [22,52,53,54]. We recently reported that CD73 molecules expressed in γδ T cells are more effective than CD73 expressed in other immune cells in converting AMP into immunosuppressive adenosine (Figure 1) and that γδ T cells express changeable levels of CD73, which correlated to converting the anti- and pro-inflammatory effects of γδ T cells [32,41]. Our studies also showed that AMP failed to inhibit the proliferation of αβ T cells in the absence of γδ T cells but was inhibitory when the responder T cells contained a small percentage (5%) of γδ T cells, and this inhibitory effect was prevented by the CD73 inhibitor—Adenosine 5′-(α,β-methylene)diphosphate (APCP) [32], suggesting that the CD73 is functionally important for γδ T cells’ regulatory function. Indeed, the CD73^+/+^, but not the CD73^−/−^, γδ T cells, have an inhibitory effect on the proliferation of αβ T cells in the presence of AMP [41].

## 5. Role of Adenosine Deaminase (ADA) in γδ T Cell Regulation

Newly formed adenosine is rapidly removed from tissues by adenosine-metabolizing enzymes. Adenosine deaminase (ADA) is an enzyme that converts adenosine into functionally inactive molecules. ADA is also an adenosine-degrading enzyme that is expressed in almost all animal tissues [55]. Exogenous ADA was initially used to treat immune deficiencies involving ADA dysfunction [11,56,57,58], but subsequent studies showed that enhanced ADA function was associated with an increased incidence of autoimmune disease [11,59] and that the suppression of aberrant ADA activity by ADA inhibitors has an anti-inflammatory effect [60,61]. Our studies showed that ADA can inhibit the development of EAU, that ADA inhibitor-erythro-9-(2-hydroxy-3-nonyl) adenine (ENHA) treatment significantly increased the EAU clinical score, serum IL-17 levels, and a percentage of IL-17^+^ αβ T cells, and that these effects were γδ dependent [62] (Figure 2).

## 6. Regulatory Effect of γδ T Cells on Foxp3^+^ Treg Cell Response

The regulatory effect of γδ T cells on the Foxp3 Treg cell response has been previously described [63,64,65,66]. While some studies have reported that γδ T cells have an inhibitory effect [63,64], the opposite effect has also been observed [14,65,66]. Our studies showed that γδ T cells can either increase or diminish Treg cell populations. The cell-free supernatants of γδ T cells had an enhancing effect on the Foxp3 T-cell response, whereas γδ T cells themselves, particularly in an activated state, strongly inhibited the Foxp3 Treg cell response [32]. We were able to show that the number of Foxp3^+^ Treg cells was elevated approximately four-fold among the CD3^+^ cells of TCR-δ^−/−^ mice compared to in B6 mice [67]; the injection of TCR-δ^−/−^ mice with a small number of activated, but not resting, γδ T cells (1 × 10^6^/mouse) before immunization significantly inhibited the Foxp3^+^ Treg cell response in vivo [67] (Figure 3). Because activated γδ T cells were better inhibitors of the Foxp3^+^ Treg cell response studied here, and because we have previously observed that activated γδ T cells expressed increased amounts of A2AR [32,41], we sought to determine whether the altered expression of A2AR contributes to this functional change. We found that A2AR^−/−^ γδ T cells were completely deficient in this suppressive activity [67], suggesting that adenosine affects γδ T cells. It is likely that preferential binding of adenosine by γδ T cells [32] diminishes adenosine binding of αβ T cells, including Foxp3^+^ Treg cells, leading to enhanced autoimmune responses. In addition, adenosine binding by γδ T cells also promotes γδ T-cell activation [32,41], leading to stronger Th17 responses.

## 7. Reciprocal Interactions between γδ T Cells and Adenosine Metabolism

γδ T cells play a major role in adenosine generation and ATP metabolism [32,41]. Adenosine is a major metabolite of ATP. In a general sense, extracellular ATP tends to be pro-inflammatory; however, when ATP is degraded into adenosine, the suppressive effect of adenosine prevails [53,54]. Inversely, a high ratio of ATP/adenosine tends to enhance the immune response [22,52]. The ecto-5-nucleotidase enzyme CD73 is pivotal in the conversion of extracellular immunostimulatory ATP (eATP) into immunosuppressive adenosine [51,52]. Although the mechanism remains unclear, we repeatedly showed that the CD73 expressed on γδ T cells was functionally more active in ATP/adenosine conversion compared to the CD73 expressed by other immune cells, such as αβ T cells, including Foxp3^+^ αβ T cells, and DCs [32,41]. In such studies, no adenosine was detectable in the supernatants of αβ or γδ T cells cultured in the absence of exogenously added AMP and adenosine generation was diminished when the CD73 inhibitor APCP was added to the cultures. However, adenosine generation was predominantly seen in the γδ T-cell cultures but not in the αβ T-cell cultures, suggesting that the CD73 on αβ T cells is essentially ineffective in the conversion of AMP into immunosuppressive adenosine, even though αβ and γδ T cells express comparable levels of CD73 [41] (Figure 1). This finding suggests that γδ T cells play a unique role in the regulation of adenosine metabolism, which may contribute to their pro-inflammatory activity. By promoting the conversion of extracellular ATP into adenosine, γδ T cells may create a microenvironment in which adenosine levels are elevated, leading to their own activation and the promotion of Th17 responses. Furthermore, our findings suggest that γδ T cells may be more effective at generating adenosine than other immune cells, which may partially explain their distinct pro-inflammatory activity. Overall, the interaction between adenosine and γδ T cells appears to be complex and bidirectional, with each promoting the activity of the other in a positive feedback loop. Further research is needed to fully understand this interaction and its role in autoimmune diseases.

## 8. Adenosine Effect on DCs

The adenosine and adenosine receptor interaction exerts numerous influences on the differentiation, maturation, and activation of cells in the mononuclear-phagocyte system [68,69]. Previous studies have shown that adenosine receptor signaling directly inhibits the effector functions of T cells and macrophages/DCs [12,70]. Our studies revealed that after treatment with adenosine, mouse bone marrow dendritic cells (BMDCs) were better able to promote Th17-autoreactive T cells, even though their ability to promote Th1 cells declined [40,71]. Likewise, when mouse BMCs were cultured in a GM-CSF-containing medium with an adenosine receptor agonist, they differentiated into a unique DC subset with greater Th17-stimulating activity [71]. We showed that adenosine-treated DCs have an increased ability to stimulate γδ T cells, and after exposure to adenosine, the ability of DCs to produce IL-12 decreased, but their ability to produce IL-23 increased, indicating that adenosine receptors are important in regulating the capability of DCs to produce factors that promote Th1 or Th17 T-cell responses. These findings suggest that the effects of adenosine on immune cells are complex. While adenosine receptor signaling can inhibit the effector functions of some immune cells, it can also enhance the stimulatory activity of DCs and promote Th17 responses in certain conditions. This result highlights the need for further research to fully understand the impact of adenosine on immune cell activation and differentiation, as well as the potential therapeutic implications of targeting adenosine receptors in different disease contexts.

## 9. Unique Role of γδ T Cells in Adenosine-Mediated Immunoregulation

Studies on the hierarchical order of adenosine capture by immune cells have offered new insights regarding the mechanisms through which adenosine regulates immune responses. Adenosine can bind to many different cell types. However, the preferential binding of adenosine to activated γδ T cells and its effect on their activation and Th17-promoting ability suggest that γδ T cells play a unique role in the regulation of adenosine-mediated immune responses. Indeed, the enhancing effect of γδ T cells on Th17 responses is based on several pathways, all of which involve adenosine:Adenosine enhances Th17 responses by promoting γδ T-cell activation. Activated γδ T cells express an increased level of high-affinity adenosine receptors (A2AR), which can lead to an “adenosine sink.” The increased absorption of adenosine by γδ T cells enhances immune responses that are otherwise inhibited by adenosine, particularly Th17 responses;Adenosine induces a functionally unique DC subset that preferentially activates Th17 cells. The influence of adenosine affects DC differentiation by favoring DCs that are capable of supporting Th17^+^ pathogenic T cells and, therefore, promotes Th17 responses, whereas Th1 responses are suppressed;Adenosine-treated DCs produce more pro-Th17 cytokines and fewer Th1-promoting cytokines. Previous studies showed that adenosine-treated DCs have a reduced ability to produce IL-12, leading to reduced Th1 responses. However, we were able to demonstrate that while the adenosine-treated DCs produced decreased amounts of IL-12, their production of IL-23 significantly increased. As a result, Th17 responses were enhanced. Moreover, adenosine can enhance γδ T-cell activation, even though its direct effect on αβ T cells is inhibitory. Consistent activation of γδ T cells is an important step leading to higher Th17 responses;Activated γδ T cells have the strongest ability to bind adenosine, and their competition with αβ T cells for adenosine diminishes the effect of adenosine on αβ T cells.

## 10. Conclusions

In summary, γδ T cells play an important role in promoting Th17 responses through their ability to bind adenosine and activate in response to it. Through its effects on γδ T cells, DCs, and cytokine production, adenosine enhances Th17 responses, which may contribute to the pathogenesis of autoimmune diseases. Therefore, targeting γδ T cells and adenosine/adenosine receptors may represent a potential therapeutic strategy for these diseases. Further studies are needed to fully understand the mechanisms by which adenosine regulates Th1 and Th17 responses and to develop effective treatments targeting this pathway.

## Figures and Tables

**Figure 1 biomolecules-13-01432-f001:**
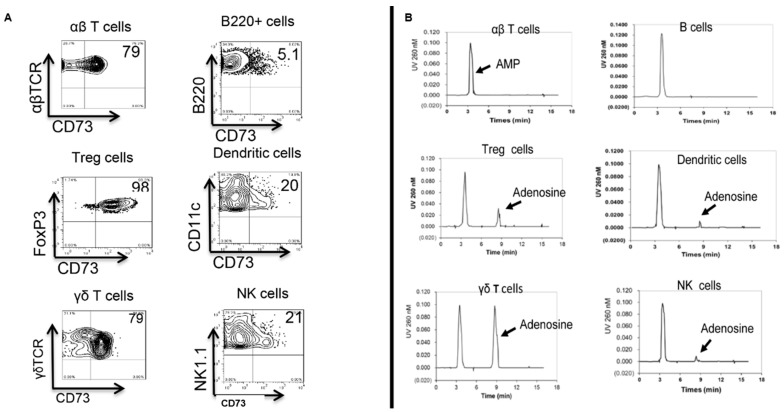
**Generation of adenosine from AMP in the supernatants of cultured immune cells.** (**A**): Flow cytometry analysis of CD73 expression by αβ T cells, B cells, FoxP3^+^ CD4^+^ Treg cells, dendritic cells, γδ T cells, and NK cells. (**B**): HPLC analysis showing the amount of adenosine in culture supernatants after 1 h culture of αβ T cells, B cells, FoxP3^+^ CD4^+^ Treg cells, dendritic cells, γδ T cells, or NK cells in the presence of 1 mM AMP. The AMP and adenosine peak is indicated by the arrows. The data are from one experiment representative of at least three independent experiments.

**Figure 2 biomolecules-13-01432-f002:**
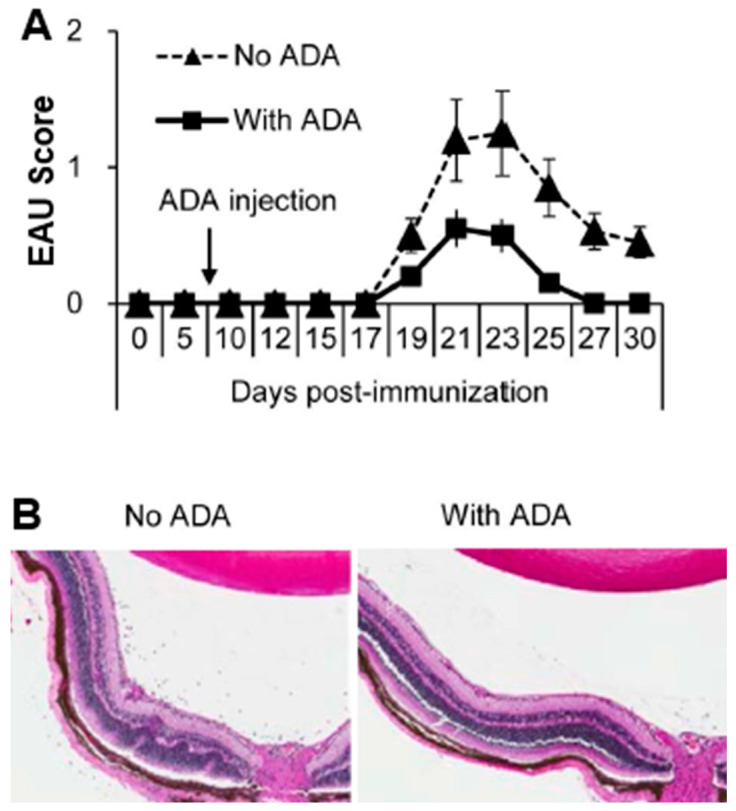
**ADA injection inhibits EAU induction in B6 mice.** Two groups of B6 mice (n = 6) were immunized with IRBP_1–20_/CFA, then, on day 8 post-immunization, one group was injected i.p. with a single dose of ADA (5U/mouse) and the other with PBS. (**A**): Clinical EAU score; (**B**): Histology.

**Figure 3 biomolecules-13-01432-f003:**
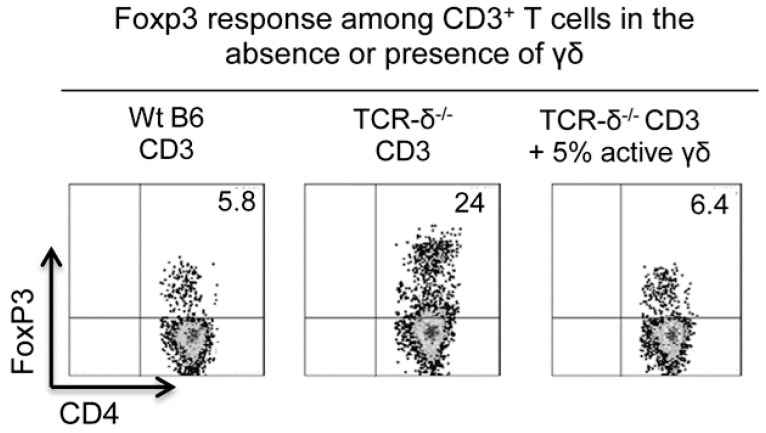
**γδ T cells possess an inhibitory effect on the Foxp3 Treg cell response.** CD3^+^ T cells (1 × 10^6^/well) prepared from immunized wt-B6 (left panel) and TCR-δ^−/−^ mice (middle and right panels) were cultured in the presence or absence of isolated γδ T cells (5 × 10^4^/well) from immunized B6. After culture in medium containing 1 ng/mL IL-2 for 5 days, the percentage of Foxp3^+^ Treg cells among CD3^+^ T cells was determined by FACS analysis. The data are from one experiment representative of at least three independent experiments.

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
