# Peer review of "The Role of Adenosine in γδ T-Cell Regulation of Th17 Responses in Experimental Autoimmune Uveitis"

_biomolecules, 2023, doi:10.3390/biom13101432_

Round 1

Reviewer 1 Report

The authors intend to show that γδ T cells negatively regulate ocular autoimmune reaction (EAU) in ocular tissues. The activities of  γδ T cellsare partially regulated by extracellular adenosine, by enhancing them to produce IFN-gamma, which can suppress Th17 reaction. In addition, the authors also demonstrate that adenosine may have different biological functions between ab and γδ T cells. This study is well designed and well organized. The data are some interesting. 

Main Concerns:

1. γδ T cells have a few subsets, which subsets responsible prefentially to adenosine need to be specified. 

Minor Concerns. 

1. The statistics treatments are required for Figure 1 and 3. 

2. CD73 FACS analysis is required for all types of cells listed in Figure 1 to demonstrate whether the expression of CD73 is involved with the concentration of AMP in their supernatant. 

3. In order to discuss the relationship between Th1 and Th17 cells, the paper: https://www.ncbi.nlm.nih.gov/pubmed/17496900, Nat Med. 2007 Jun;13(6):711-8. doi: 10.1038/nm1585. Epub 2007 May 13., should be citated for discussion. 

The English is ok.

Author Response

  1. “γδ T cells have a few subsets, which subsets responsible preferentially to adenosine need to be specified.”

-In this study γδ T cell is mainly activated in the periphery. Vγ1+ and Vγ4+ are the major subsets of the peripheral γδ T cells. Both subsets respond to adenosine. Since in immunized mice Vγ4+ γδ T cells are the dominant subset and as we previously reported that Vγ1+ and Vγ4+ T cells have distinct activation requirements, a greater effect of adenosine on γδ T cells in immunized mice should be contributed by the Vγ4+ cells. We have added this paragraph in Page 6 Line 9.

  1. “The statistics treatments are required for Figure 1 and 3. CD73 FACS analysis is required for all types of cells listed in Figure 1 to demonstrate whether the expression of CD73 is involved with the concentration of AMP in their supernatant.”

-We have revised Fig.1 and added CD73 expression by different immune cells analyzed by FACS. Fig.1 and Fig.3 are results from one experiment representative of at least three independent experiments (see Page 14).

  1. “Citations of Th17 paper.”

-The paper recommended by the reviewer has been cited as reference 7.

Reviewer 2 Report

The authors present a review on the role of adenosine in Th17 regulation in experimental autoimmune uveitis through interaction with gamma delta T cells

 Comments:

1)      The abbreviation "Th1" should be explained as "T-helper 1 (Th1)"

2)      Th17 cells were discovered in 2005. I would write "more recently" rather than "recently" in the introduction section.

3)     They respond to the same treatment. The sentence is not clear, it is better to explain what kind of treatment is meant (immunosuppressive, anti-inflammatory...).

4)      In the introduction, the authors should better clarify the purpose of the review (e.g., "The purpose of this review is..."

5)      The major problem with this review is that the authors discuss their personal studies too extensively This should be corrected so that the review does not appear to be a discussion of an original article.

English requires moderate editing

Author Response

  1. “The abbreviation "Th1" should be explained as "T-helper 1 (Th1)”

-The change has been made in Page 2 Line 1.

  1. “ Th17 cells were discovered in 2005. I would write "more recently" rather than "recently" in the introduction section”

-The change has been made in Page 4 Line 13.

  1. “The major problem with this review is that the authors discuss their personal studies too extensively This should be corrected so that the review does not appear to be a discussion of an original article.”

-The major points of this review paper is to point out how different Th1 and Th17 pathogenic T cells respond to the regulatory effects of γδ T cells and extracellular adenosine. Since the emphasis of this review has been placed on the role of adenosine in γδ T cell activation on which our laboratory has done the most of the mentioned experiments, the scope of the discussion has not included many other important topics related to structure and function of γδ T cells.

Round 2

Reviewer 1 Report

The manuscript has been significantly improved after being successfully revised and I recommend it for publication in your journal.

Reviewer 2 Report

The authors fully responded to my comments and modified significantly the manuscript in accordance with the remarks